# Targeting KRAS Regulation with PolyPurine Reverse Hoogsteen Oligonucleotides

**DOI:** 10.3390/ijms23042097

**Published:** 2022-02-14

**Authors:** Alexandra Maria Psaras, Simonas Valiuska, Véronique Noé, Carlos J. Ciudad, Tracy A. Brooks

**Affiliations:** 1Department of Pharmaceutical Sciences, School of Pharmacy and Pharmaceutical Sciences, Binghamton University, Binghamton, NY 13902, USA; apsaras@binghamton.edu; 2Department of Biochemistry and Physiology, School of Pharmacy and Food Sciences, & IN2UB, University of Barcelona, 08028 Barcelona, Spain; simonasvaliuska13@gmail.com (S.V.); vnoe@ub.edu (V.N.); cciudad@ub.edu (C.J.C.)

**Keywords:** KRAS, PPRH, G-quadruplex, pancreatic cancer, ovarian cancer

## Abstract

KRAS is a GTPase involved in the proliferation signaling of several growth factors. The KRAS gene is GC-rich, containing regions with known and putative G-quadruplex (G4) forming regions. Within the middle of the G-rich proximal promoter, stabilization of the physiologically active G4_mid_ structure downregulates transcription of KRAS; the function and formation of other G4s within the gene are unknown. Herein we identify three putative G4-forming sequences (G4FS) within the *KRAS* gene, explore their G4 formation, and develop oligonucleotides targeting these three regions and the G4_mid_ forming sequence. We tested Polypurine Reverse Hoogsteen hairpins (PPRHs) for their effects on *KRAS* regulation via enhancing G4 formation or displacing G-rich DNA strands, downregulating *KRAS* transcription and mediating an anti-proliferative effect. Five PPRH were designed, two against the *KRAS* promoter G4_mid_ and three others against putative G4FS in the distal promoter, intron 1 and exon 5. PPRH binding was confirmed by gel electrophoresis. The effect on *KRAS* transcription was examined by luciferase, FRET Melt^2^, qRT-PCR. Cytotoxicity was evaluated in pancreatic and ovarian cancer cells. PPRHs decreased activity of a luciferase construct driven by the *KRAS* promoter. PPRH selectively suppressed proliferation in KRAS dependent cancer cells. PPRH demonstrated synergistic activity with a *KRAS* promoter selective G4-stabilizing compound, NSC 317605, in KRAS-dependent pancreatic cells. PPRHs selectively stabilize G4 formation within the KRAS mid promoter region and represent an innovative approach to both G4-stabilization and to *KRAS* modulation with potential for development into novel therapeutics.

## 1. Introduction

KRAS is a 21-kD GTPase that plays a role in cell survival, proliferation, and differentiation [1,2]. It is constitutively expressed, but active only when GTP-bound. Normal functioning KRAS has a relatively short, and inducible, GTP-bound life. Mutations in RAS proteins are found in approximately one-third of all human tumors, with KRAS being the most frequently mutated isoform [3,4]. Single point mutations of the *KRAS* gene abolish inherent GTP hydrolysis; these mutations render the protein constitutively active. The highest incidence of mutational activation occurs in lung, colorectal, and >95% of pancreatic cancers [1,4,5]. KRAS mutations are associated with increased tumorigenicity and poor prognosis [3]. Mutation of the *KRAS* gene has been identified as a transforming oncogenic event, where it creates an unstable environment allowing for more mutational selection and increasingly aggressive disease [5]. In the absence of a mutation, increased KRAS activity in human tumors is the result of gene amplification, overexpression, or increased upstream activation [3]. The genomic amplification of KRAS, in particular, is associated with metastatic disease and poor prognosis in hormone-related cancers such as ovarian cancer [6,7,8,9,10,11,12,13,14,15,16,17,18,19,20,21].

KRAS is a well validated anti-cancer therapeutic target. Many active drug discovery programs are ongoing, with a focus on individual mutant KRAS isoforms. Sotorasib targets the G12C mutant protein and was FDA-approved for lung cancer in late May of 2021 [22,23]. While this drug’s development, clinical activity, and approval is remarkable, G12C is not a common mutation in pancreatic and colorectal cancers with KRAS mutations and sotorasib does not benefit patients with amplified or overexpressed KRAS. The approach of developing agents for each KRAS mutation necessitates an array of drugs to be developed and does not address cancers with dysregulation of non-mutant KRAS. Notably, transcriptional down-regulation has been demonstrated to be lethal to tumor cells with aberrant KRAS signaling, irrespective of mutational status, and to potentially have a wide therapeutic window [1,24,25,26,27]. Stabilization of higher order genomic structures is an established approach to modulating transcription.

Within the proximal promoter region of KRAS lies a GC-rich region of DNA capable of forming non-canonical G-quadruplex (G4) structures. G4s are secondary structures made of four guanine bases associated by Hoogsteen hydrogen bonds and a stack of a minimum two of these structures forming helical composition. G4 structures have an important role in controlling different biological processes such as DNA replication [28] telomere homeostasis [29] and mRNA transcription and regulation, processing and translation [30,31]. Since they are present in biological key functions, these formations can be considered as potential therapeutic targets. G4 formation in both DNA and RNA has been demonstrated in live cells [28] in an inducible manner [32]. Moreover, G4-positive nuclei are significantly increased in cancer cells, as compared to surrounding non-neoplastic tissue, from patient-derived solid tumor tissue biopsies [33]. G4s are attractive therapeutic targets as they are more globular than B-DNA, enabling more selective gene interactions. The core promoter region of *KRAS* is highly G/C-rich (~75%) and putatively capable of forming higher order non-B-DNA structures [34,35,36]. The proximal promoter region contains several G4-forming sequences, including G4_near_ (32r) and G4_mid_. Previous work from our collaborative group described the predominant G4 isoform formed by G4_mid_, and ascribed G4-mediated silencing within this region to G4_mid_ formation [37]. Several other putative G4-forming regions exist within the promoter and the gene itself whose formation and function have yet to be described.

In this work we targeted the G-rich, putative G4-forming regions with PolyPurine Reverse Hoogsteen hairpins (PPRHs), which are two polypurine strands linked by a four-thymidine loop (4T) running in antiparallel directions and bound by intramolecular Hoogsteen bonds, forming a hairpin structure. These molecules can hybridize in a specific sequence-dependent manner to single- or double-stranded DNA (dsDNA) by Watson-Crick bonds forming a triplex structure and displacing the fourth strand of the dsDNA which can lead to a knockdown of the targeted gene [38]. We classify the PPRHs depending on the strand of the DNA they target. If the polypyrimidine stretch is found in the template strand they are called Template-PPRHs [39] and if the stretch is found in the coding strand, they are called Coding-PPRHs [40]. The latter can also target mRNA, where it can modulate post-transcriptional events, since it has the same orientation as the coding strand. PPRHs have already been shown to inhibit gene expression of cancer related genes [41] in immunotherapy [42,43,44] or in replication stress [45]. In addition, they can be used for gene repair to correct point mutations [46,47] and very recently have also been used as the sensor component of biodetectors [48,49].

Previously, we observed that targeting the complementary strand of G4 forming sequences (G4FS) in the 5′-UTR of the thymidylate synthase (TYMS) gene was very effective in reducing the viability, mRNA, and protein levels of this target [50]. Here we designed and used PPRHs against polypyrimidine tracts in the promoter and gene coding region of *KRAS*, making them complementary to G4FS to promote possible G4 folding and related gene silencing or post-transcriptional modifications. We targeted the promoter region of this gene, including proximal and distal segments and an intronic part within the 5′-UTR. Additionally, we designed a PPRH targeting exon 5 in the 3′-UTR of the gene.

## 2. Results

### 2.1. Identification of Putative PPRHs Target Sequences in KRAS Promoter and Gene Region

G4′s formed within the proximal KRAS promoter have been previously described [37,51], including the physiologically active KRAS G4_mid_ sequence. We included this region and searched for other G4FS within the *KRAS* gene sequence using both the TFO searching tool and the QGRS mapper (Figure 1A,B), and then selected sequences that could form triplexes with the highest G-Score. Notably, the QGRS identified portions of the G4_mid_ sequence with G-scores of 20–21; inputting only the G4_mid_ sequence containing seven G-tracts, the mapper identified a wide array of putative G4s with G-scores ranging up to 39. The predicted G4 with the highest G-score encompasses the second (termed B) through fifth (termed E) G-tracts, in agreement with the structure identified previously [37], and is targeted by two of the PPRHs designed below. We further designed the PPRHs targeting the polypyrimidine stretches complementary to the known G4_mid_ or the G4FS (Figure 1C). The KRAS sequence with the designed hairpin target regions is shown in Figure 1D. Figure 1E represents PPRHs mechanism of action with the example of HpKRasPrEF-C (PPRH 1).

### 2.2. G4 Formation within Newly Identified G4FS–PR, I1 and E5

While the G4 formation within the proximal promoter region has been previously explored and described, it is unknown for the newly identified sequences within the distal promoter (KRasPr-C, PR), intron 1 (KRasI1-T, I1) and exon 5 (KRasE5-C, E5). Using electronic circular dichroism, we examined the induction of G4 sequences upon the addition of 100 mM KCl (Figure 2A). The distal promoter sequence did not demonstrate any secondary structure in either the absence or presence of KCl, whereas the intron 1 sequence formed a structure without variation in the absence or presence of KCl that is consistent with a hairpin loop. Only the sequence within exon 5 demonstrated the induction of a parallel G4 upon the addition of KCl. All sequences were melted from 20–100 °C; mdeg were recorded every 1 °C at 262 nm and full spectra were recorded every 10 °C (Appendix A). With no apparent formation of a secondary structure within the PR region, no melting profile was able to be described. Both the I1 and the E5 sequences, however, were probed without and with 100 mM KCl to examine their thermal stability (T_M_). The T_M_ of I1 increased from 46 to 51 °C upon the addition of KCl, whereas the T_M_ of E5 markedly increased from 45 to 75 °C under the same conditions (Figure 2A, right). The inter- versus intra-molecular tendencies of the E5 G4 was explored through the examination of dose dependent G4 formation and thermal stability in the presence of 100 mM KCl (Figure 2B). G4 formation and thermal stability increased in a dose dependent manner, with maximal mdeg at 264 nM increasing from 1 to 14 theta and T_M_s increasing from 58 to 70 °C, consistent with an interstrand G4.

### 2.3. PPRHs Binding to KRAS Target Sequences and Polupurine Strand Displacement

To evaluate the binding of the designed PPRHs to their corresponding target regions, we performed different electrophoretic mobility shift assays (EMSA; Figure 3) in a native gel testing the PPRHs with dsDNA of KRAS target sequences (Table 1). In the case of PPRH 1 and HpKRasBC-C (PPRH 2) we designed a sole dsDNA target probe spanning the entire KRAS G4_mid_ sequence, encompassing the binding sites for both PPRHs. However, for PR, I1 and E5, we designed a dsDNA target probe corresponding to their sequences. For each of the PPRHs tested, we observed one main shifted band that corresponds to the binding of the hairpin with the probe. In contrast, the negative control, Hp-Sc9 (SCR), did not show any shifted band for any of the probes.

In order to assess whether the binding of the targeting PPRHs mediated the displacement of the G-rich strand, we performed a strand displacement assay using incubations of the dsDNA-KRAS-I1 probe with increasing concentrations of HpKRAS-I1 and staining with thioflavin T (ThT) [52,53] after resolving the structures by native gel electrophoresis (Figure 3B). The Hp-KRAS-I1 was able to displace the polypurine strand in a concentration-dependent manner.

A single polypurine strand, and Hp-KRAS-I1, are able to form non-canonical structures, such as a G4 [52] or hairpin [53], since these bands stained with ThT presenting a cyan band. Combination of dsDNA-KRAS-I1 with Hp-KRAS-I1, led to a formation of several bands, the upper corresponds to the intron 1 dsDNA-PPRH triplex, the second upper band corresponds to HpKRAS-I1 excess, the third band coincides with dsDNA-KRAS-I1 as we observe a decrease of this band with the increase of HpKRAS-I1. The lower band corresponds with the displaced ssPPU band of the triplex formed as we observe a higher cyan fluorescence of this band with a higher amount of PPRH per dsDNA. Incubation of dsDNA-KRAS-I1 with the scrambled PPRH (Hp-Sc9), lane 8, did not show any displaced band. The independency of KCl for structured formations in the intron 1 probe is shown in (Appendix A), in agreement with the ECD results; similar displacement results for the distal promoter probe were also noted (Appendix A). Notably, all PPRH sequences were examined by ECD for their secondary structure formation (Appendix A). In the absence of KCl, sequences were either single-stranded or noted to form hairpin structures (data not shown). Except for the scrambled sequence (HpSc9), they each formed structures that facilitate ThT binding as noted in Figure 3. Specifically, PPRH1, PPRH2, and Hp-KRAS-E5-C formed inducible G4 structures, Hp-KRAS-PR-‘s maxima at 266 nm sits somewhere in between a G4 and a hairpin and Hp-KRAS-I1-T forms an apparent hairpin structure.

### 2.4. Effect of KRAS-Targeting PPRHs on Cell Viability and Growth

We examined the dose-dependent effects of KRAS-targeting PPRHs (144 h) on the viability of KRAS mutant expression pancreatic cancer cell lines AsPc-1 (G12D, highly addicted to KRAS) and MiaPaCa-2 (G12C, moderately addicted to KRAS) and KRAS overexpressing ovarian cancer SKOV-3 cell lines (Figure 4A). All effects on viability were compared to untreated control, and vehicle (N-[1-(1,2-Di-(9Z-octadecenoyl)-3-trimethylammoniumpropane methyl sulfate, DOTAP) and scrambled (Sc9, SCR) sequences were included as negative controls. All KRAS-targeting PPRHs significantly decreased the viability of AsPc-1 cells in a dose-dependent manner, but surprisingly none had lone efficacy in the moderately addicted MiaPaCa-2 cells. Targeting the promoter region with PPRH 1 or HpKRAS-PR-C (PR), or intron 1 with HpKRAS-I1-T (I1), significantly decreased cell viability in SKOV-3 cells as well. We further explored the effects of each PPRH at 100 nM over time in the three cell lines using live cell microscopy over time and evaluating percent confluency in the wells. In accordance with the viability effects, all the PPRHs modulated the growth of AsPc-1 cells and targeting the promoter with PPRH 1 or PR had the most marked effects on SKOV-3 growth. The effect of DOTAP on the growth of SKOV-3 cells within 16–96 h occurred consistently (*n* = 3 with 25 images taken per well every 8 h) and normalized after 96 h; cells incubated with scramble DNA in DOTAP did not exert any stimulation of growth. Interestingly, targeting the promoter also decreased the growth of MiaPaCa-2 cells, despite there being no significant change in the viability of the remaining cells. One key difference in the viability and the cell growth studies is the well size (96 well versus 6 well), so we also monitored viability in the larger plates used for cell growth. In the 6 well plate, PPRH 1 mediated a decrease in MiaPaCa-2 viability by 35%, and PR decreases viability by 13%; while the magnitude and significance does not match the findings with cell growth alone, the general trends are in agreement. Overall, targeting transcription of KRAS with PPRHs targeting either the distal or proximal promoter region modulated the growth of cells with mutant or overexpressed KRAS.

To assess the possible additive or synergistic effects of combining PPRHs targeting different parts of the KRAS gene, we examined the effects of PPRH 2 (50 nM) plus each other PPRH (50 nM). DOTAP and SCR DNA (50 and 100 nM) were used as comparative controls. SCR DNA (50 nM) was also combined with either PPRH1 or PPRH2 (50 nM) and the effects on cell viability were comparable to PPRH1 or PPRH2 at 50 nM alone (data not shown). All other combinations of SCR + PPRH at 50 nM each were not examined, but were rather compared to 100 nM of SCR DNA alone. PPRH2 was chosen for all combinations as targeting the KRAS G4_mid_ region was efficacious at decreasing cell viability and growth in the cell lines tested; it demonstrated promoter-related decreases in activity (see below), and its activity was moderate in all cell lines, allowing for enhancement or inhibition to be noted. One-way ANOVAs with Tukey post-hoc analyses were utilized to evaluate effects that were more than their constitutive parts, and all were compared to the negative controls of SCR DNA or DOTAP alone (Figure 5). While significant effects were noted with 100 nM of combined effects in SKOV-3 cells as compared to either negative control, the effects were apparently additive. No combinatorial effects were noted in the MiaPaCa-2 cells. Notably, in the cell line which has been reported to be the most sensitive to KRAS modulation, AsPc-1, apparent synergistic effects were noted with most PPRH combinations. PPRH 2 at 50 nM alone decreased viability of AsPc-1 cells by 14%, and the other agents decreased viability by 13–28%, whereas all combinations containing PPRH 2 decreased cell viability by 73–88%.

### 2.5. Combined Effect of Targeting KRAS Promoter G4s with PPRH and Small Molecules

Finally, we sought to understand the effect of the combined targeting of the KRAS promoter G4 with PPRHs and small molecules. As only PPRH 1 and PPRH 2 target the confirmed G4 within the KRAS gene, this final study focused only on this region and these PPRHs. The regulation of KRAS promoter activity by PPRH 1 and PPRH 2 (100 nM, each) was first examined utilizing the KRAS promoter-driven luciferase cassette previously described [37] as compared to a promoterless empty vector (EV). Both PPRH moieties decreased KRAS promoter activity in a time-dependent manner, with PPRH 2 demonstrating significant decreases within 48 h of transfection.

NSC 317605 (hereafter referred to as 317605) is an indoloquinoline compound identified through screening the NCI DTP Diversity Set III’s 1600 compounds by the FRET Melt^2^ assay [54] for stabilizers of the KRAS G4_mid_ structure. The thermal stability of the KRAS G4_mid_ structure was increased by 8 °C, whereas there was no effect on the KRAS G4_near_ structure (ΔT_M_ < 0.5 °C) (Figure 6B, left). This is notable as both G4s have been described in the literature; however, the biologically silencing function of G4-modulation on KRAS has been identified within the mid region and notably the mid region is the target of the designed PPRH 1 and PPRH 2 oligonucleotides. 317605′s activity on the other KRAS sequences described in the current work, in the presence of KCl (100 mM), was also examined and there was no induction of a G4 structure within the PR or I1 sequences. E5′s G4 was retained, but there was no increase in G4 formation as evidenced by increased theta, and there was no increase in thermal stability (ΔT_M_ = 0.5 °C) (data not shown).

In cells, 317605 (1 μM) significantly decreased KRAS promoter activity within 48 h of transfection and treatment (Figure 6B, middle). In vitro potency of this compound is moderate, with a 72 h IC_50_ in the AsPc-1 cell line of 21 μM, but KRAS expression is significantly decreased (by 50%) at the [IC_50_]. As PPRH 1 and PPRH 2 both target the same G4-forming region stabilized by 317605, combinations of SCR, PPRH 1 or PPRH 2 and the small molecule were examined in the AsPc-1 cell line (Figure 6C). Both PPRH 1 and PPRH 2, but not SCR, dose-dependently enhanced the cytotoxic effects of 317605 at concentrations up to 50 nM. Notably, as described above, these are concentrations of the PPRH moieties that are not markedly cytotoxic in the AsPc-1 cells, highlighting the synergistic activities of the small molecule/PPRH approach to KRAS modulation.

## 3. Discussion

In the present study, we sought to develop a novel means of modulating KRAS expression and/or function by targeting gene regulation with PolyPurine Reverse Hoogsteen hairpins (PPRH) targeting putative G4-forming DNA sequences within the KRAS gene. PPRHs are novel therapeutic tools that can be used for gene silencing [55], gene repair [46,47] to validate new targets and for diagnostic purposes [48,49]. PPRHs target polypyrimidine tracts, and in this work, we selected stretches of polypyrimidines that had putative G4 forming sequence (G4FS) in the complementary strand. PPRHs have multiple applications and exhibit several advantages. Since these hairpins are non-modified oligonucleotides, they are very economical to synthesize. Due to the hairpin conformation made of deoxynucleotides, PPRHs present high stability and, because their length is less than 100 nucleotides, the absence of immunogenicity [56]. Additionally, these moieties inhibit gene activity at low concentrations, and they are sequence specific. Recently, we described the use of a PPRH targeting the complementary strand of a G4FS in the 5′UTR region of the Thymidylate synthase which produced a great reduction in viability, mRNA, and protein levels in PC3 and HeLa cells [50]. Following the same philosophy, in the present work we designed 5 PPRHs–4 coding and 1 template–directed against various G4FS in the *KRAS* gene.

We identified and examined higher order DNA formation in three previously unexplored regions of *KRAS*–one distal promoter, one intronic and one exonic. Of those, we identified one region capable of forming interstrand G4 structures which have the potential to form in the pre-transcriptionally modified mRNA. PPRHs were further designed to target these three novel G-rich DNA regions independent of their G4-forming capabilities, in addition to the previously identified and described G4-forming region within the proximal promoter, G4_mid_ [54]. The PPRH oligonucleotides interacted with their target DNA in a sequence-specific manner. Since the binding of PPRHs to their targets provokes strand displacement, the aim of our approach was to either facilitate G4 formation or to displace the G-rich strand to disrupt transcription factor binding. Independent of G4 formation, G-rich DNA strand displacement could inhibit transcription to downregulate gene expression or alter post-transcriptional modification to interfere with functional KRAS protein expression, either of which would decrease proliferation of KRAS dependent ovarian and pancreatic cancer cells. We found that PPRHs have varying efficacy in pancreatic and ovarian cancer cells, but consistently that targeting the promoter with PPRHs inhibits cellular growth and viability. The PPRHs demonstrated synergistic activity in AsPc-1 cells, and additivity in SKOV3 cells. We further identified and described a G4_mid_-selective small molecule, NSC 317605, and demonstrated the synergistic activities with enhanced selectivity for the *KRAS* promoter G4_mid_, improved transcriptional downregulation and correlating effects on KRAS-dependent AsPc-1 pancreatic cancer cells. This combinatorial approach has the potential to enhance the selectivity and activity of G4-stabilizing small molecules and is a novel means to G4-stabilization and gene control.

G4s are globular constructs that form from a single strand of guanine-rich DNA that folds on itself to form a higher order, non-canonical, four-stranded structure. DNA intramolecular G4 formation occurs in telomeres, promoter regions, at origins of replication, regions of DNA breaks and at locations of DNA repair [28,57,58,59,60]. G4 formation within promoter regions requires opening of the double-stranded (ds) DNA, which occurs under the negative torsional stress induced by transcription. Genes with higher transcription rates have a greater potential for promoter G4 formation [61]. This underscores the role of most promoter G4s as negative regulators of gene expression, as their formation mediates a local negative feedback loop that regulates transcription rates. G4 formation in both DNA and RNA has been demonstrated in live cells [28] in an inducible manner [32]. Moreover, G4-positive nuclei are significantly increased in cancer cells, as compared to the surrounding non-neoplastic tissue, from patient-derived solid tumor tissue biopsies [33]. G4 formation is also inducible by small molecules, and endogenous levels of G4s can predict tumor sensitivity to G4-targeted ligands [62].

Within the core of *KRAS* promoter (+50 to −510 bp from the transcription start site) it has been reported that there are transcription factor (TF) binding sites for Sp1, E2F-1, STAT4, MAZ, WT1 or P53. Notably, the target sequences of the PPRHs contain binding sites of TFs such as Sp1 and E2F-1 and therefore the hairpins could impair the interaction between the TFs and the promoter independent of G4 formation, thus decreasing transcription [54]. The results obtained in the luciferase experiments corroborate that there was indeed a decrease in transcription provoked by the promoter targeting PPRHs, alone and in combination with the small molecule, indoloquinoline NSC 317605.

Pancreatic cancer overall, and aggressive ovarian cancers in particular, harbor aberrant KRAS signaling correlating with more metastatic and progressive disease and poorer response to chemotherapies [6,17,63,64,65]. Pancreatic cancer is the ninth most common cancer in terms of incidence, and the fourth most common cause of cancer-related deaths in the United States. While ovarian cancer incidence is less common, it is the third most common genital system malignancy and the fifth most common cause of cancer related deaths in women. Decreased KRAS expression is a validated therapeutic approach for these and other cancers harboring addictions to aberrant KRAS signaling [1,9,26,28]. Stabilization of higher order genomic structures–such as the G4–is an established approach to modulating transcription. The work described herein identifies and combines two therapeutic strategies–PPRHs and the small molecule NSC 317605–capable of stabilizing the *KRAS* promoter G4_mid_ structure, decreasing transcription and synergistically modulating KRAS addicted AsPc-1 pancreatic cancer cells.

PPRHs, a new type of therapeutic oligonucleotide, are DNA molecules composed of two symmetrical stretches of polypurines separated and connected by four thymidines, forming a hairpin structure wherein Hoogsteen base-pairing facilitates G:G and A:A bonds. These structures further utilize Watson-Crick base-pairing to form triplex structures with cytosine-rich regions of DNA, freeing the guanine-rich complementary strand to facilitate G4 formation, displace TF binding, and modulate post-transcriptional events [41,66,67,68,69,70]. PPRH ODNs binding to the mid-G4-forming region of the *KRAS* promoter enhance G4 structures in vitro, mediating KRAS downregulation and subsequent cytotoxicity. Moreover, PPRH ODNs demonstrate synergistic activity with small molecules in KRAS-dependent pancreatic cell lines. The binding of PPRHs to their targets is sequence-specific and decreases transcriptional activity, viability and confluency. Therapeutic development for selective stabilization of individual promoter G4s is an active area of research, and our approach of combining PPRH moieties with small molecules to stabilize particular promoter G4s is novel and has strong potential for enhanced selectivity and anti-cancer activity.

## 4. Materials and Methods

### 4.1. Design of Polypurine Reverse Hoogsteen Hairpins

Putative *KRAS* targets of PPRH hairpins were searched using the Triplex-Forming Oligonucleotide Target Sequence Search software (TFO) (Triplex-Forming Oligonucleotide Target Sequence Search. Available online: http://utw10685.utweb.utexas.edu/tfo/ (accessed on 10 February 2021)), MD Anderson Cancer Center, University of Texas, Houston, TX, USA). From a list of putative *KRAS* targets in the TFO searching tool, we selected the polypurine stretches that follow the best criteria according to our previous results: a minimum of 40% G, a length between 20 and 25 nucleotides per strand of PPRH and allowing no more than three pyrimidine interruptions.

Within the selected KRAS PPRHs hairpins target sequences we searched for putative G4FS using the QGRS mapper (http://bioinformatics.ramapo.edu/QGRS/index.php (accessed on 10 February 2021)), a computational tool that uses algorithms to map putative G-quadruplex elements in mammalian genes. Next, we selected the sequences with the highest G-score that represents the best candidates to form G4FS. To confirm the specificity of the designed PPRHs and avoid unintended targets, the final candidate sequences were analyzed using BLAST.

The final design of the PPRH hairpins consists of two polypurine strands linked by a four-thymidine loop in a mirror repeat fashion, thus running in antiparallel orientations. As negative control we designed a scramble hairpin (Hp-Sc9). All the designed sequences were synthetized as non-modified oligodeoxynucleotides by Sigma-Aldrich (Haverhill, UK) resuspended in sterile Tris-EDTA buffer (10 mM Tris and 1 mM EDTA, pH 8.0) (Sigma-Aldrich, Madrid, Spain) and stored at −20 °C.

### 4.2. Electronic Circular Dichroism (ECD)

DNA sequences (Table 1) were purchased from Integrated DNA technologies (IDT, Coralville, IA, USA) as desalted oligonucleotides. Upon arrival, they were solvated in double-distilled water overnight, they were heated to 95 °C for 5 min and their A260 was determined at temp using a Nanodrop3000 (Thermo Scientific, Waltham, MA, USA) and their concentrations were determined using the nearest neighbor technique. On the experimental day, oligonucleotides were diluted to ~5 (range 2–20) mM in 10 mM Tris Acetate buffer (pH 7.4), in the absence or presence of 100 mM KCl; experimental concentrations were confirmed as described above by heating to 95 °C for 5 min and measuring the A260 at temperature, DNA was rapidly cooled on ice for 10 min. Spectra and thermal stability of the putative G4 forming regions were evaluated on a Jasco J-1500 spectrophotometer (Jasco, Easton, MD, USA). Spectra were recorded from 225–350 nm in triplicate for each experiment using a 1 cm quartz cuvette and a 1 mm bandwidth; the triplicate reads were averaged. Full spectra were recorded over increasing temperature from 20–100 °C, with recordings of the Cotton effect at 262 nm at each temperature. Millidegrees (mdeg, theta) were reported as experimentally determined, or normalized for the thermal melt determinations; T_M_s were calculated using the data at 262 nm over temperature using non-linear regression analysis with GraphPad Prism software (San Diego, CA, USA).

### 4.3. Electrophoretic Mobility Shift Assay (EMSA)

To perform EMSA analyses, we used fluorescently labeled dsDNA probes, corresponding to the target regions of each PPRH (Table 2), obtained by hybridizing (95 °C for 5 min and cool down at RT) equimolecular amounts of single-stranded oligodeoxynucleotide (Table 2) in a 150 mM NaCl solution. The probes were labeled in the 5′-end with 6-FAM (fluorescein) in the polypyrimidine ssDNA and were synthesized by Sigma-Aldrich (Haverhill, UK). Binding reactions were performed using binding buffer (5% glycerol, 36 mM KCl, 25 mM Tris-HCl, 4 mM MgCl_2_, 0.5 mM DTT, 0.5 mM EDTA, pH 8.0; all reagents were from Sigma-Aldrich). The different PPRHs (1 µg) were mixed with 200 ng of Poly(dI:dC) as a nonspecific competitor and incubated at 65 °C for 10 min. Afterwards, 200 ng of dsDNA probe were added to the mix at 65 °C for an additional period of time of 20 min. The products of the binding were electrophoretically resolved in 7% polyacrylamide and 5% glycerol native gels in 0.5× TBE buffer, at a fixed of 190 V and 4 °C. ImageLab software v5.2 was used to visualize the results (GE Healthcare, Barcelona, Spain).

### 4.4. Strand Displacement Assay upon PPRH Incubation

To detect G4 Structures, 1.5 μg of each oligonucleotide alone (Table 2) or in combination with the indicated amounts of PPRH were mixed in 60µL of buffer containing 100 mM Tris-HCl (pH 7.4) and 100 mM KCl and then incubated at 90 °C for 5 min in water and slowly cooled down to room temperature (90 min). dsDNA probes were prepared by hybridizing the two strands where the polypyrimidine one was labeled with FAM following the same protocol as in 4.3 of M & M. The resulting structures were resolved in native polyacrylamide gels (12%) containing 10 mM KCl in 1× TBE buffer and electrophoresed for 1–2 h at 150 V. After electrophoresis, gels were stained with 5 μM of Thyoflavin T solution for 15 min under agitation and washed in water for 2 min. Images were captured under a UV light lamp or using the Gel DocTM EZ with the Image Lab Software, Version 6.0.

### 4.5. Cell Cultures

All cell lines were purchased from American Tissue Culture Collection (Manassas, VA, USA) and stored in liquid nitrogen until use. Low passage cells (<20) were maintained for the duration of these experiments. Pancreatic cancer AsPc-1 and MiaPaCa-2 cell lines were grown in RPMI 1640 and Dulbecco’s Modified Eagle’s Medium (DMEM) media, respectively, ovarian cancer SKOV-3 cells were maintained in McCoy’s 5A media, and HEK293 cells were maintained in Eagle’s Minimum Essential Media (EMEM); media were supplemented with 10% fetal bovine serum (Sigma Aldrich, St Louis, MO, USA) and penicillin/streptomycin (Sigma Aldrich). All cells were maintained in exponential growth at 37 °C in a humidified 5% CO_2_ incubator.

### 4.6. Cellular Viability and Cell Growth Studies

One day before transfection, cells were seeded in 96 or 12 well plates at 0.5–2.5 × 10^3^ and 8–20 × 10^4^ cells per well in 90 or 900 mL of corresponding media, respectively. PPRHs were incubated with N-[1-(1,2-Di-(9Z-octadecenoyl)-3-trimethylammoniumpropane methyl sulfate (DOTAP; Sigma Aldrich) in a 1:100 ratio at 10x solutions in OptiMEM media for 20 min at room temperature to form micelles. Micelles or compounds were diluted over a 5–6 log range in OptiMEM media and 10 or 100 mL were added to the 96- or 6-well cell plates, respectively. Cells were incubated with the PPRHs for up to 144 h, or with NSC317605 for 72 or 144 h, at 37 °C in a humidified 5% CO_2_ incubator. The combined effect of PPRHs and NSC317605 was examined by exposing the cells to a 5–6 log range of NSC317605 in the absence or presence of 0–50 nM of PPRH’s. Only 5 mL of the PPRH of compound were added to the plated cells, thus the stocks were 20x rather than the 10x described above. To determine effects on cellular viability, Cell Titer AQeuous (MTS) reagent (Promega; Madison, WI, USA) was activated with 5% phenazine methosulfate (Sigma Aldrich), and 20 or 200 mL of the activated reagent was added to the 96- or 12-well plates and incubated for 2–4 h. Absorbance was measured at 490 nm on a SpectraMax i3x (Molecular Devices; San Jose, CA, USA). Background absorbance (media and all reagents) was subtracted from all experimental values and normalized to untreated controls. Non-linear regression was performed with GraphPad Prism software for the dose-response studies, and a two-way ANOVA with a post-hoc Tukey analysis was utilized to evaluate statistical significance for all experimental groups. Additionally, live-cell images were captured of all 12-well plates every 8 h after transfection utilizing the CellCyte X Live imaging system (Cytena; Boston, MA, USA). Twenty-five images per well were captured every 8 h; analysis software was trained to accurately determine the shape and volume of each cell line, and the “masks” created by training were applied to determine percent confluency within each images’ surface area. Gompertzian growth was analyzed by GraphPad Prism and two-way ANOVAs with Tukey post hoc analyses was performed. Cell viability and cell confluency/growth studies were all performed in triplicate.

### 4.7. FRET Melt^2^

5′FAM and 3′TAMRA labeled G4_mid_ (5′-GGCGGGGAGAAGGAGG*GGG*CC*GGG*CCGGGCCGGC*GGG*GGAGGAGC*GGG*GGCCGGGCCG-3′) or G4_near_ (5′-TGA*GGG*C*GG*T*G*T*GGG*AAGAGGGAAGAG*GGG*GAGG-3′) DNA (consecutive guanines underlined, guanines involved in resolved predominant G4 isoforms italicized [38]) was diluted to 0.5 mM in 10 mM sodium cacodylate (pH 7.4) with 90 mM LiCl, 10 mM KCl and 10% glycerol [54]. The G4 was annealed by heating the DNA to 95 °C for 5 min and rapidly cooled on ice for 10 min, twice. Annealing was done in either the absence or presence of 2 mM of NSC317605. Fluorescence was then recorded from 20–95 °C, at every degree with a hold for 10 s on a Bio-Rad CFX96 real-time PCR machine (BioRad Laboratories; Hercules, CA, USA). Fluorescence for each sample at each temperature was normalized to the values determined at 20 °C, and non-linear regression was performed using GraphPad Prism to determine the T_M_. Experiments were performed in triplicate with internal duplicates.

### 4.8. Luciferase Assays

HEK-293 cells were stably transfected with either the pGL4.17 promoterless luciferase plasmid (Promega) or the KRAS promoter containing KRAS-324 luciferase plasmid (FL; [37]), selected by neomycin resistance. HEK-293-EV or -FL cells were seeded in 24 well plates at 8 × 10^4^ cells/well and allowed to attach overnight. Cells were treated with 100 nM PPRH, 1 mM NSC317605, or vehicle control (DOTAP or DMSO), for 48 h. Cells were lysed in passive lysis buffer, frozen to −20 °C, thawed, and refrozen before measuring firefly luciferase activity with the Dual Luciferase Assay kit (Promega) using a Lumat LB9507 luminometer. The Pierce BCA assay (Thermo Fisher) was used to determine the protein concentration from each sample, and luciferase was normalized to protein content, and normalized again to untreated control. Luciferase assays were performed minimally in triplicate; one-way ANOVAs with Tukey post-doc analyses were used to determine significance.

### 4.9. RT-qPCR

To determine *KRAS* mRNA levels in AsPc-1 cells at the [IC_50_], cells were plated at 3–4 × 10^5^ cells/well in a 6 well plate and allowed to attach overnight. Cells were treated with 21 mM NSC 317605 for 72 h, lysed and RNA was harvested with the Roche GeneJet RNA isolation kit (Thermo Fisher). 500 ng of RNA was reversed transcribed into cDNA, and 1/5 of the resultant cDNA was mixed with 1× TaqMan PerfeCTa qPCR SuperMix (QuantaBio; Beverly, MA, USA), 1 μL of FAM-labeled *KRAS* probe (ABI, Thermo Fisher Hs00364282_m1 and 1 μL of primer-limited VIC-labeled *GAPDH* probe (ABI, Thermo Fisher Hs02758991_g1). PCR cycling conditions were 3 min denaturation at 95 °C, followed by 40 cycles of 10 s at 95 °C and 30 s at 60 °C. Relative expression of *KRAS* was determined using the DDC_T_ method from regression modeled C_T_ values; experiments were performed in triplicate with minimally duplicate qPCR samples. One-way ANOVA with post-hoc Tukey analysis was used to determine statistical significance.

### 4.10. Statistical Analyses

Statistical analyses were carried out using GraphPad Prism 9 (GraphPad Software, San Diego, CA, USA) using the tests described in the corresponding text. All data, with a minimum of three independent experiments, are shown as the mean ± SEM.

## Figures and Tables

**Figure 1 ijms-23-02097-f001:**
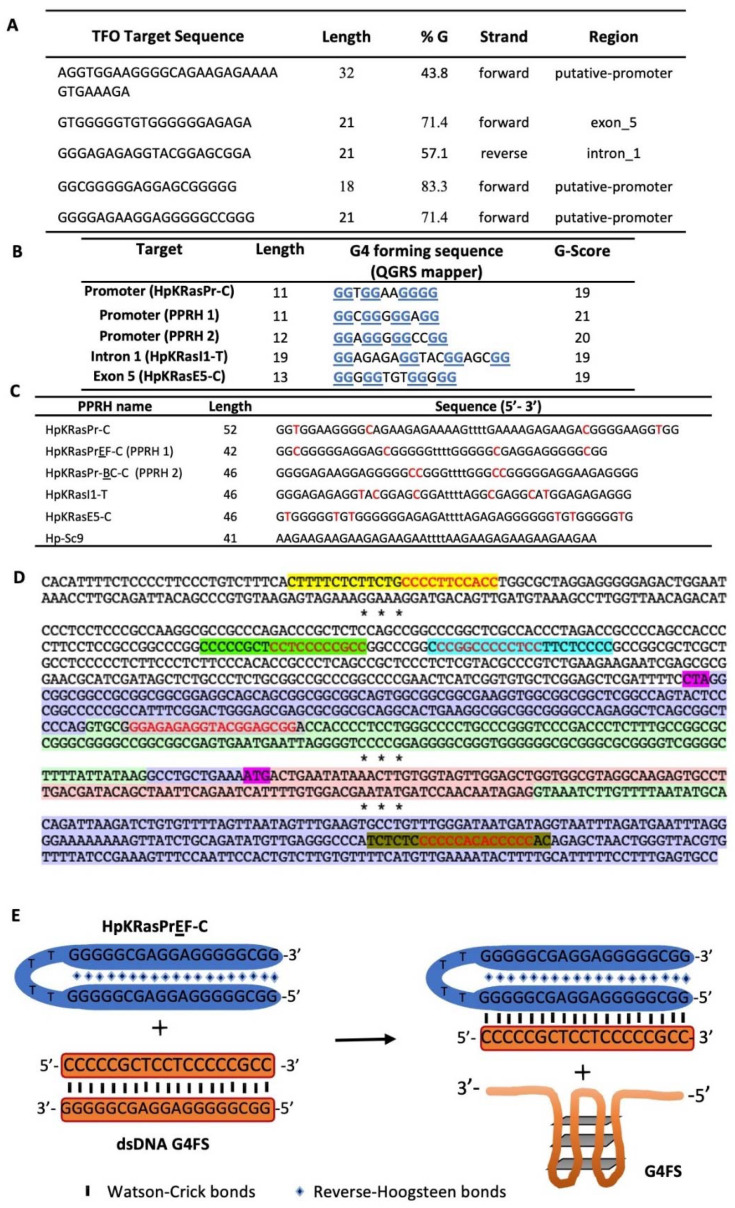
(**A**) Putative target regions of K-Ras using the TFO searching tool and searching the sequence in the Promoter Database. (**B**) QGRS mapper results of the selected sequences. Guanines marked in blue and underlined are involved in G4 formation. (**C**) Design of the PPRHs targeting K-Ras specific G4FS (HpKRasPr-C, HpKRasPrEF-C, HpKRasPrBC-C, HpKRasI1-T, HpKRasE5-C) and a scramble hairpin, Hp-Sc9 as a negative control. Pyrimidine interruptions are marked in red. (**D**) Location of the target regions of the selected PPRHs: four of them belong to the coding category (HpKRasPr-C highlighted in yellow, HpKRasPr-EF-C in light green, HpKRasPr-BC-C in cyan, and HpKRas-E5-C in brown) and HpKRas-I1-T in grey which is a template-PPRH. The putative G4FS, or its complementary sequence, are marked in red. Three PPRHs have targets in the promoter sequence (white zone). Another PPRH binds to intron 1 (green) which is within the 5′-UTR (purple). The beginning of transcription (CTA) and translation (ATG) are indicated in magenta, right before the first coding region (pink). The last PPRH is designed towards a sequence within the 3′-UTR in Exon 5. Asterisks indicate gaps in the sequence of K-Ras. (**E**) Putative mechanism of PPRH 1 (HpKRasPrEF-C) strand displacement of the dsDNA G4FS target sequence, facilitating G4-formation and subsequent gene regulation.

**Figure 2 ijms-23-02097-f002:**
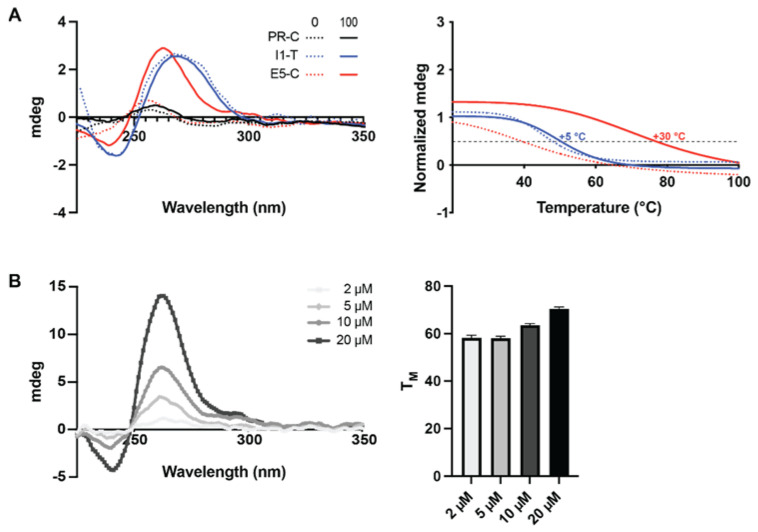
G4 formation within newly identified G4FS. (**A**) Indicated DNA strands were annealed in the absence (dotted lines) or presence (solid lines) of 100 mM KCl and their spectra were recorded from 225–350 nm (left). Thermal stability of the I1 and E5 structures were determined in the absence (dotted lines) and presence (solid lines) of 100 mM KCl (right). Only the E5 sequence demonstrated induction of a G4 structure (with parallel topology) with increased thermal stability in the presence of KCl. (**B**) E5′s G4 formation (left) and thermal stability (right) were further examined over a range of concentrations (2–20 M) in the presence of 100 mM KCl as an indicator of inter- versus intra-strand G4 formation. Both G4 formation and thermal stability increased as a function of concentration, indicating inter-strand G4 formation.

**Figure 3 ijms-23-02097-f003:**
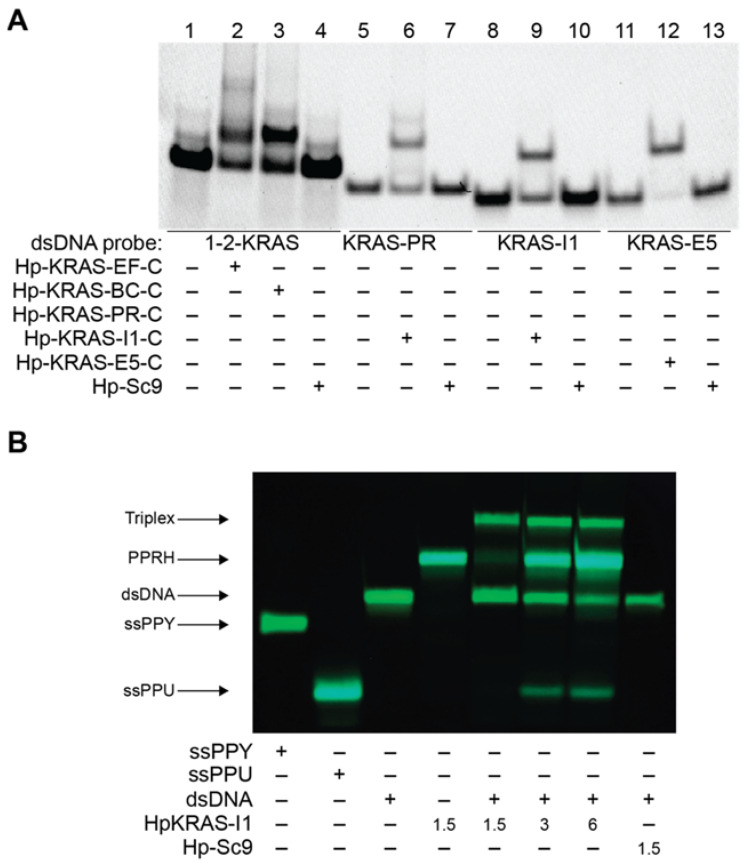
(**A**) Bindings of K-RAS PPRHs and Hp-Sc9 (1µg) to the complementary polypyrimidine G4FS target sequence dsDNA (200 ng) marked with FAM. Lane 1, dsDNA-1-2-KRAS probe alone; lane 2, dsDNA-1-2-KRAS probe plus HpKRasPrEF-C; lane 3, dsDNA-1-2-KRAS probe plus HpKRasPrBC-C; lane 4 dsDNA-1-2-KRAS probe plus Hp-Sc9; lane 5, dsDNA-KRAS-PR-UP probe alone; lane 6, dsDNA-KRAS-PR-UP probe plus HpKRAS-PR-C; lane 7, dsDNA-KRAS-PR-UP probe plus Hp-Sc9; lane 8, dsDNA-KRAS-I1 probe alone; lane 9, dsDNA-KRAS-I1 probe plus Hp-KRAS-I1-T; lane 10, dsDNA-KRAS-I1 probe plus Hp-Sc9; lane 11, dsDNA-KRAS-E5 probe alone; lane 12, dsDNA-KRAS-E5 probe plus Hp-KRAS-E5-C; lane 13, dsDNA-KRAS-E5 probe plus Hp-Sc9. The image is representative of at least three different EMSAs performed at different times. (**B**) Displacement analysis of the Polypurine (PPU) strand in Intron 1 probe. Bindings were performed using 1.5 µg of dsDNA labelled with FAM (green) in the polypyrimidine (PPY) strand only, then incubated with the indicated amounts of KRAS-I1 PPRH or 1.5 µg of the negative control HpSc9. The resulting structures were resolved by native polyacrylamide (12%) gel electrophoresis. PPRHs, ssPPU and displaced PPU were visualized after Thioflavin-T staining (cyan bands).

**Figure 4 ijms-23-02097-f004:**
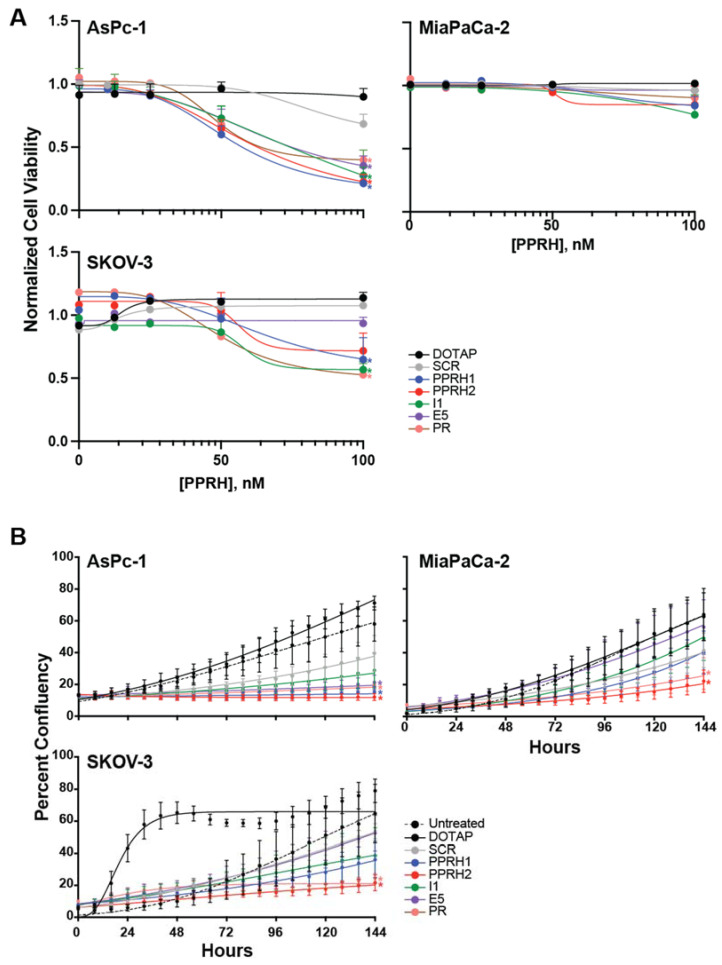
Effect of KRAS-targeting PPRHs on pancreatic cell lines expressing mutant KRAS, AsPc-1 and MiaPaCa-2, and ovarian cancer KRAS overexpressing SKOV-3 cells. (**A**) All cells were incubated with a dose-range (up to 100 nM) of the indicated sequences for 144 h. (**B**) The effects of 100 nM of each sequence on cancer cell line growth was measured overtime. Color coded * *p* < 0.05 versus DOTAP and SCR controls as determined by two-way ANOVA with Tukey post-hoc testing, all experiments were performed minimally in triplicate with internal triplicate data for the cellular viability studies and 25 images per time point for the live cell microscopy cellular growth studies.

**Figure 5 ijms-23-02097-f005:**
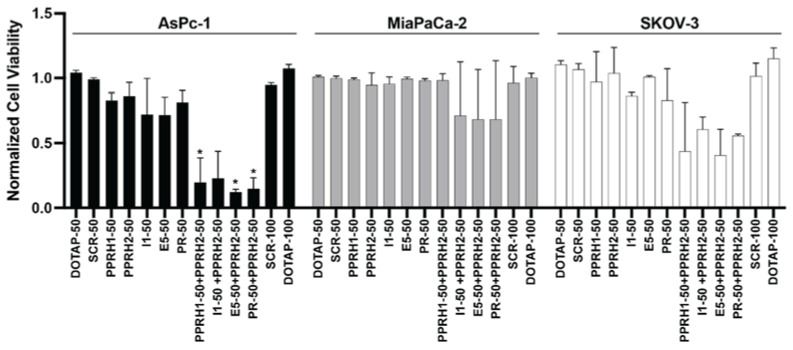
Combinations of PPRHs targeting the KRAS gene in AsPc-1 and MiaPaCa-2 pancreatic cancer cells and ovarian cancer SKOV-3 cells. Each PPRH was examined alone or in combination at 50 nM in the indicated cell lines for 144 h, and changes in cell viability were noted. Vehicle (DOTAP) and scramble DNA (SCR) controls were examined at both 50 and 100 nM. * *p* < 0.05 in the combinations, as compared to either component alone, as determined by one-way ANOVA with Tukey post-hoc testing. All experiments were performed minimally in triplicate with internal triplicate data.

**Figure 6 ijms-23-02097-f006:**
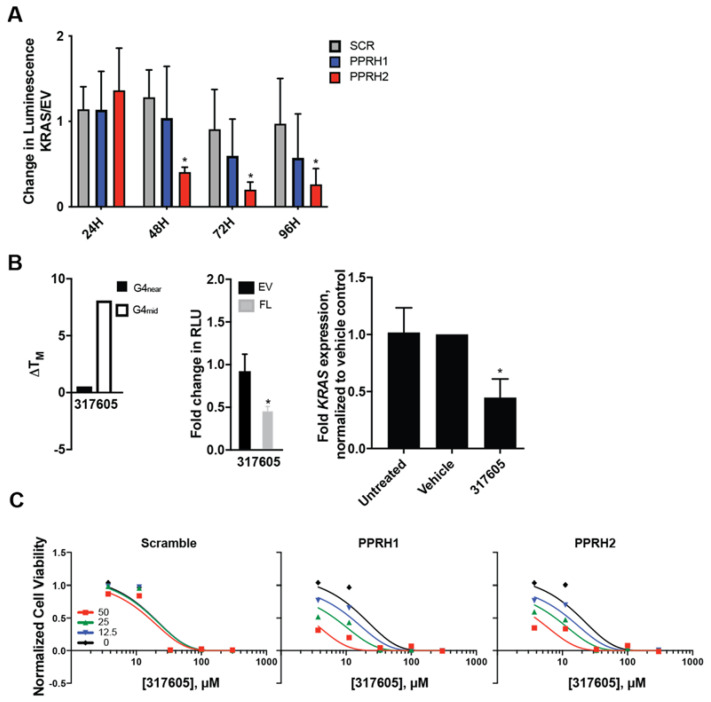
Combinatorial effects of targeting the KRAS promoter G4_mid_ structure with small molecules and PPRHs. (**A**) PPRHs significantly decreased KRAS promoter activity within 48 h, and extended through 96 h in HEK-293 cells stably transfected with the KRAS FL promoter, but not the promoterless EV. (**B**) 317605 was identified as a KRAS G4_mid_ selective stabilizer, versus G4_near_, from the NCI DTP Diversity Set III. FRET Melt^2^ (left) demonstrated a marked increase in thermal stability of G4_mid_ with correlations in decreased KRAS promoter activity (middle) and KRAS gene expression in AsPc-1 cells at the 72 h [IC_50_]. (**C**) AsPc-1 cells were exposed to dose-ranges of both 317605 and the designated PPRHs for 144 h. PPRH1 and PPRH2, but not scramble oligonucleotides, sensitize the pancreatic cancer cells to 317605, showing a synergistic effect. * *p* < 0.05 as compared to relevant scramble or untreated controls. All experiments were completed minimally in duplicate, with internal replicates as indicated in the materials and methods.

**Table 1 ijms-23-02097-t001:** Name and nucleotide sequences used in the Electronic Circular Dichroism assays.

Name	Sequence (5′-3′)
I1	CGGGAGAGAGGTACGGAGCGGAC
E5	GTGGGGGTGTGGGGGGA
PR	CAGGTGGAAGGGGCA

**Table 2 ijms-23-02097-t002:** Name and sequence of the DNA probes used in the EMSA.

Name	Sequence (5′-3′)
Probe-1-2-PY-KRAS	5′-[6FAM]-CCCCCGCTCCTCCCCCGCCGGCCCGGCCCGGCCCCCTCCTTCTCCCC-3′
Probe 1-PPU-KRAS	5′-GGGGAGAAGGAGGGGGCCGGGCCGG-3’
Probe 2-PPU-KRAS	5′-GCCGGCGGGGGAGGAGCGGGGG-3′
Probe-PPY-KRAS-PR-UP	5′-[6FAM]-CTTTTCTCTTCTGCCCCTTCCACC-3′
Probe-PPU-KRAS-PR-UP	5′-GGTGGAAGGGGCAGAAGAGAAAAG-3′
Probe-PPY-KRAS-I1	5′-[6FAM]-TCCGCTCCGTACCTCTCTCCC-3′
Probe-PPU-KRAS-I1	5′-GGGAGAGAGGTACGGAGCGGA-3′
Probe-PPY-KRAS-E5	5′-[6FAM]-TCTCTCCCCCCACACCCCCAC-3′
Probe-PPU-KRAS-E5	5′-GTGGGGGTGTGGGGGGAGAGA-3′

## Data Availability

Not applicable.

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
