# Peer review of "Targeting KRAS Regulation with PolyPurine Reverse Hoogsteen Oligonucleotides"

_ijms, 2022, doi:10.3390/ijms23042097_

Round 1

Reviewer 1 Report

Overall Comment:

This study explores the targeting of the KRAS oncogene through G-quadruplex structures that form within its promoter region using synthesized oligomers of polypurine reverse hoogsteen (PPRH) hairpins. The Authors provide a solid rationale for examining KRAS in appropriate model systems of prostate and ovarian cancers. The findings reveal a newly identified KRAS G4 structure and support a new, viable alternative approach to targeting structures with PPRH and thus, contributes to both DNA secondary structure and KRAS oncology fields. As such, the manuscript is recommended for publication following a few additional statements for clarification and minor revisions to further strengthen the work.

Major:

  1. Authors conclude from findings presented in Figure 2 that the G4 sequence within the distal region and intron do not form stable G4s, but continued to test the PPRHs designed against these sequences. And in most cases the intron PPRH had one of the greater cytotoxic effects (viability) and the distal PPRH for growth. Is the G4 sequence dramatically induced to fold into a G4 with the PPRH, despite the inability for KCl to induce formation? Can the Authors comment on what they think is underlying this efficacy? Perhaps the comment in the Discussion regarding transcription factor binding can be directly associated with these PPRHs and expanded upon.
  2. The Tm melt curves for the new distal G4, PR, in Figure 2A, right panel seem to be missing.
  3. In Figure 3, the Authors nicely demonstrate binding of the designed PPRH to the given KRAS dsDNA version of the G4 sequence with inclusion of the negative control of a scrambled PPRH. Did the Authors examine whether the PPRHs interacted with the other G4 sequences, eg. Hp-KRAS-EF-C with dsDNA-KRAS-E5, etc.?
    1. Did Authors determine induction/stabilization of G4 formation in addition to binding?
  1. For combination experiments presented in Figure 5, would an alternative to the 100 nM SCR-PPRH control be, 50 nM-SCR + 50 nM-of a given PPRH? This combination will most likely produce similar effects on cell viability as the 50 nM-of a given PPRH, but this Reviewer was interested in the choice for 100 nM SCR-PPRH as a control in addition to the 50 nM SCR-PPRH only.

  1. Does the Tukey post-hoc analysis distinguish between additive or synergistic effects, or simply correct for multi-hypothesis testing and determine significance for an effect?
  2. Why would the Authors expect synergy between PPRH1 and PPRH2 – would they potentially outcompete each other? Why was PPRH1 not used in combination with the exon 5 PPRH? Combinations seem to only be conducted with PPRH2.
  3. Similar to the first part of the question above, although more likely that compound and PPRHs bind differently to the G4, how does the compound stabilizer synergize with the PPRHs, do they not displace one another, bind to different positions on the G4s, does the stabilizer help hold G4 in confirmation for the PPRH to then bind better to opposite strand and maintain strand displacement? Could 317605 also stabilize the newly identified G4s?

Minor:

  1. Typo in abstract, line 17 “strangs”, did Authors mean “strands”?
  2. Figure 2B is missing a y-axis title on left panel graph and legend states mM as concentration not microM as in the figure legend.
  3. Figure 2 is included twice, along with identical Results sections 2.2 and 2.3, but this glitch may have been a pdf processing issue?
  4. DOTAP is not defined at first use.
  5. Typo in Results, page 9, line 235 – “whereas” requires capitalization. Same with the last sentence in the legend for Figure 5.
  6. Unclear use of word “surround” in Discussion, page 11, lines 333-334
  7. In Materials and Methods, housekeeping gene is offered as primer-limited probe from Taqman not a primer-delimited (line 500).

Reviewer 2 Report

The manuscript by Psaras et al reports an interesting finding by the authors on KRAS-targeting PPRHs. The study appears to be well designed, executed, and presented. I have no objection to having the manuscript published in IJMS as it provides us with sufficient quality. A few minor suggestions to improve the manuscript are as follows:

  1. (Line 12) The authors must explain what G4mid stand for, for it appears for the first time.
  2. (Line 104) What is ref Cogo, Morgan? It should be cited as the other references.

Author Response

Thank you for identifying the corrections needed.  Both items have been corrected.